# Single-cell transcriptome analysis of cavernous tissues reveals the key roles of pericytes in diabetic erectile dysfunction

Seo-Gyeong Bae[1†], Guo Nan Yin[2†], Jiyeon Ock[2†], Jun-Kyu Suh[2]*, Ji-Kan Ryu[2,3]*, Jihwan Park[1]*

[1]School of Life Sciences, Gwangju Institute of Science and Technology (GIST), Gwangju, Republic of Korea; [2]National Research Center for Sexual Medicine and Department of Urolog, Inha University School of Medicine, Incheon, Republic of Korea; [3]Program in Biomedical Science & Engineering, Inha University, Incheon, Republic of Korea

*For correspondence:
jksuh@inha.ac.kr (J-KyuS);
rjk0929@inha.ac.kr (J-KR);
jihwan.park@gist.ac.kr (JP)

[†]These authors contributed equally to this work

**Abstract** Erectile dysfunction (ED) affects a significant proportion of men aged 40–70 and is caused by cavernous tissue dysfunction. Presently, the most common treatment for ED is phosphodiesterase 5 inhibitors; however, this is less effective in patients with severe vascular disease such as diabetic ED. Therefore, there is a need for development of new treatment, which requires a better understanding of the cavernous microenvironment and cell-cell communications under diabetic condition. Pericytes are vital in penile erection; however, their dysfunction due to diabetes remains unclear. In this study, we performed single-cell RNA sequencing to understand the cellular landscape of cavernous tissues and cell type-specific transcriptional changes in diabetic ED. We found a decreased expression of genes associated with collagen or extracellular matrix organization and angiogenesis in diabetic fibroblasts, chondrocytes, myofibroblasts, valve-related lymphatic endothelial cells, and pericytes. Moreover, the newly identified pericyte-specific marker, Limb Bud-Heart (Lbh), in mouse and human cavernous tissues, clearly distinguishing pericytes from smooth muscle cells. Cell-cell interaction analysis revealed that pericytes are involved in angiogenesis, adhesion, and migration by communicating with other cell types in the corpus cavernosum; however, these interactions were highly reduced under diabetic conditions. Lbh expression is low in diabetic pericytes, and overexpression of LBH prevents erectile function by regulating neurovascular regeneration. Furthermore, the LBH-interacting proteins (Crystallin Alpha B and Vimentin) were identified in mouse cavernous pericytes through LC-MS/MS analysis, indicating that their interactions were critical for maintaining pericyte function. Thus, our study reveals novel targets and insights into the pathogenesis of ED in patients with diabetes.

## eLife assessment

The authors provide **important** insights into the pathogenesis of erectile dysfunction (ED) in patients with diabetes. The authors present **compelling** evidence, using single-cell transcriptomic analysis in both mouse and human cavernous tissues, to support their claims regarding the key roles of pericytes in diabetic ED. The identification of LBH as a potential pericyte-specific marker in both mouse and human tissues further strengthens their findings. This well-written manuscript offers novel and significant contributions to the field, identifying potential therapeutic targets for further investigation.

## Introduction

Erectile dysfunction (ED) is an age-dependent vascular disease, affecting 5–35% of men aged 40–70 in varying degrees (*Kubin et al., 2003*). Although not fatal, 322 million men worldwide will be affected by the disease by 2025 (*Ayta et al., 1999*; *Kessler et al., 2019*). Moreover, it occurs in up to 75% of diabetes cases (*Sáenz de Tejada et al., 2002*). This causes significant physical and psychological problems for patients and their families (*Bivalacqua et al., 2003*). Moreover, the cavernous tissue is more prone to this disease than other blood vessels; therefore, its malfunction is considered as a biomarker in the progression of cardiovascular and vascular diseases (*Miner, 2011*). Current ED treatment involves the use of phosphodiesterase 5 (PDE5) inhibitors, but patients with severe vascular disease, such as diabetic ED, are less responsive. This is mainly because severe vascular dysfunction caused by diabetes results in insufficient bioavailable nitric oxide (NO) for PDE5 inhibitors to induce penile erection (*Angulo et al., 2010*). Therefore, it is necessary to identify novel therapeutic targets, which requires a comprehensive understanding of the cavernous microenvironment, including cell-cell interactions, regulatory signals, and molecular regulation. Although recent studies have proposed gene expression profiles of primary cultured cavernous cells exposed to hyperglycemic conditions (*Anita et al., 2022*; *Ghatak et al., 2022*; *Yin et al., 2021*), the relevant cell-specific relationships and genetic mechanisms in cavernous tissues have not been elucidated yet.

Pericytes are vital in the pathogenesis of erectile function as their interactions with endothelial cells are essential for penile erection. Pericytes interact with various cell types (especially endothelial cells) and are involved in angiogenesis, vasoconstriction, and permeability (*Bergers and Song, 2005*; *Gerhardt and Betsholtz, 2003*). Diabetes causes pericyte loss or dysfunction, thereby creating an imbalance between pericytes and endothelial cells, leading to vascular diseases (*Liu et al., 2021*; *Warmke et al., 2016*). For example, diabetic retinopathy is characterized by pericyte loss, capillary basement membrane thickening, and increased permeability, leading to retinal hypoxia and inflammation. Pericytes are particularly sensitive to glucose oxidation, which further promotes pericyte apoptosis (*Warmke et al., 2016*). Advanced glycation end products (AGEs) accumulate in pericytes under hyperglycemic conditions, thereby stimulating the secretion of transforming growth factor beta in peripheral nerve pericytes, leading to basement membrane thickening, neovascularization, and pericyte apoptosis (*Shimizu et al., 2011*). Furthermore, our previous studies showed that restoring the content and function of cavernous pericytes significantly reduced the permeability of cavernous blood vessels and enhanced neurovascular regeneration (*Yin et al., 2015*; *Yin et al., 2020*; *Yin et al., 2021*). However, the mechanisms underlying the effect of diabetes on pericyte dysfunction in the ED remain unclear.

Smooth muscle cells (SMCs) are also a vital component of the cavernous tissues in penile erection. Diabetes mellitus (DM) leads to SMC dysfunction, such as decreased cavernosal smooth muscle relaxation, resulting in an inability to obtain or maintain satisfactory erection (*Reddy et al., 2016*; *Wei et al., 2012*). Since the marker genes of pericytes and SMCs often overlap, and their expression patterns are tissue-specific, identifying suitable in vivo markers to accurately interpret transcriptional and phenotypic changes in these cell types is of utmost importance (*Chasseigneaux et al., 2018*; *Kumar et al., 2017*; *Smyth et al., 2018*). Recent studies have identified pericyte-specific markers in different tissues using single-cell RNA sequencing (*Baek et al., 2022*; *Sun et al., 2022*), enabling characterization of various cell types and their cell-cell communication. In this study, a single-cell transcriptome analysis was performed to examine the cellular heterogeneity in ED caused by diabetes. The goal of this study was to identify markers specific to cavernous pericytes and understand their role in ED by exploring their interactions with other cell types.

## Results

### Single-cell transcriptional landscape of mouse cavernous tissue

To elucidate the cellular landscape of mouse cavernous tissue and transcriptional changes in diabetic ED at the cellular level, we conducted a single-cell transcriptome analysis, profiling 12,894 cells (*Figure 1A*). The cells were grouped into 15 clusters based on their transcriptomic patterns, and cell types were annotated based on the expression of marker genes (*Figure 1B and C*). Clustering analysis identified fibroblasts (FBs), chondrocytes (Chonds), myofibroblasts (MFBs), lymphatic endothelial cells (LEC), valve-related LEC, vascular endothelial cells (VEC), SMCs, pericytes, Schwann cells,

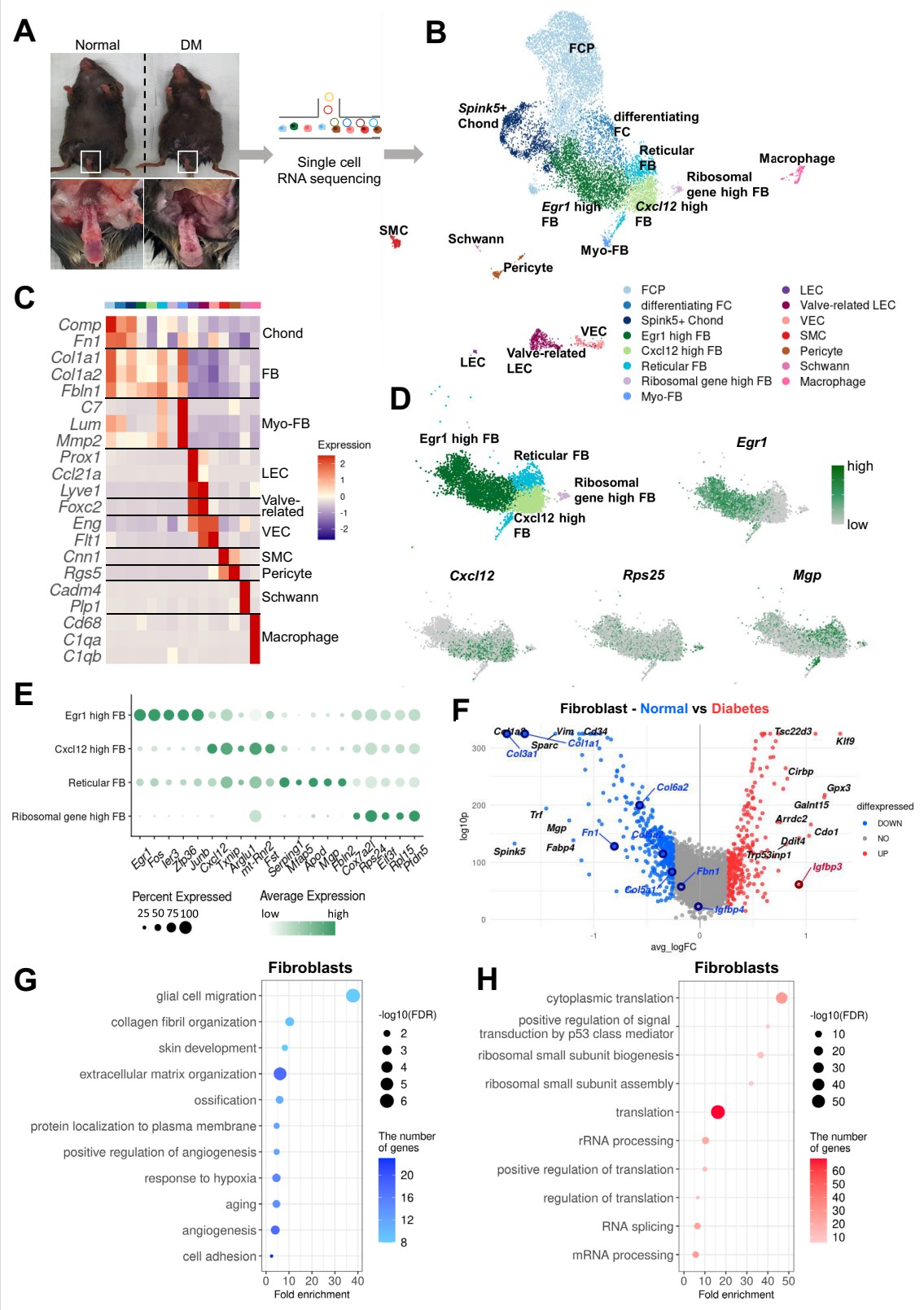

**Figure 1.** Single-cell transcriptional landscape of mouse cavernous tissues in normal and diabetic conditions. (**A**) Schematic workflow of this study. Cavernous tissues from 16-week-old male mice were used for single-cell RNA sequencing (n=5 for each group). DM, diabetes mellitus. (**B**) Visualization of single-cell data from mouse cavernous tissues using Uniform Manifold Approximation and Projection (UMAP). Each cell type is indicated by a different color. FCP, fibrochondrocyte progenitors; FC, fibrochondrocytes; Chonds, chondrocytes; FBs, fibroblasts; MFBs, myofibroblasts; LEC, lymphatic

*Figure 1 continued on next page*

*Figure 1 continued*

endothelial cells; VEC, vascular endothelial cells; SMCs, smooth muscle cells. (**C**) Heatmap of known cell-type marker genes used for annotation. (**D**) UMAP of four fibroblast clusters and expression of marker genes of four fibroblast subsets. *Egr1* for *Egr1* high FB; *Cxcl12* for *Cxcl12* high FB; *Mgp* for reticular FB; *Rps25* and *Rps17* for ribosomal gene high FB. (**E**) Dot plot showing the expression of top five marker genes of each fibroblast subset. (**F**) Differentially expressed genes (DEGs) between diabetic and normal conditions in fibroblasts. The top 10 (based on log-fold change) DEGs are indicated with gene names, and genes identified as having high or low expression in diabetes in previous studies are indicated with gene names in red or blue. DEGs with adjusted p-value>0.05 were indicated in gray. (**G**) Gene ontology analysis of the DEGs higher in normal compared to diabetes in fibroblasts. (**H**) Gene ontology analysis of the DEGs higher in diabetes compared to normal in fibroblasts.

The online version of this article includes the following figure supplement(s) for figure 1:

**Figure supplement 1.** Chondrocyte subsets in single-cell RNA sequencing data of mouse cavernous tissues.

**Figure supplement 2.** Volcano plots showing differentially expressed genes (DEGs) between diabetic and normal conditions in each cell type.

**Figure supplement 3.** Gene ontology high in normal compared to diabetes in each cell type.

**Figure supplement 4.** Gene ontology high in diabetes compared to normal in each cell type.

and macrophages. The Chond and FB subsets were further annotated with additional marker genes (*Figure 1D and E* and *Figure 1—figure supplement 1*).

Furthermore, we analyzed cell type-specific transcriptional changes due to diabetes (*Figure 1F* and *Figure 1—figure supplement 2A–G*). We found that *Nos3* (highly expressed in the untreated diabetes group compared to the treatment group) decreased in diabetic valve-related LEC and VEC (*Ismail et al., 2020*). In addition, *Igfbp2* and *Igfbp4* (lowly expressed in the penis of diabetic rats) were also downregulated in diabetic SMCs and Chonds, respectively, whereas *Igfbp3* was upregulated in diabetic FB and MFB (*Abdelbaky et al., 1998*). Collagen genes (vital in the corpus cavernosum structure) were significantly underexpressed in diabetic FB, Chond, My-FB, and pericytes than in normal tissues (*Sullivan et al., 2005*). Gene ontology analysis using differentially expressed genes (DEGs) indicated that the genes associated with collagen or extracellular matrix organization and angiogenesis were downregulated in diabetic FB, Chonds, MFBs, valve-related LEC, and pericytes compared to the cells under normal conditions (*Figure 1G* and *Figure 1—figure supplement 3*). In contrast, diabetic mice showed an increase in ribosome- and translation-related terms (*Figure 1H* and *Figure 1—figure supplement 4*). In Schwann cells, there were no significant DEGs between diabetic and normal cells.

## Identifying specific markers for pericytes in mouse cavernosum tissues

We distinguished SMC from pericytes by examining the expression of previously known marker genes (*Cnn1, Acta2, Myh11, Tagln,* and *Actg2* for SMC, *Rgs5, Pdgfrb, Cspg4, Kcnj8, Higd1b,* and *Cox4i2* for pericytes) (*Figure 2A* and *Figure 2—figure supplement 1A*). However, rather than being exclusively expressed in each cell type, known marker genes were co-expressed in each cell type. To ensure proper annotation of SMC and pericytes, we identified the gene sets enriched in each cluster through gene set enrichment analysis (GSEA). We found that the gene sets matching pericyte and SMC functions were enriched in each cluster (*Figure 2B*). Gene sets related to the regulation of blood vessels and leukocytes were enriched in pericytes, whereas gene sets related to actin or muscles were enriched in SMC, suggesting that these clusters were properly annotated. In addition, we screened six genes (*Lbh, Ednra, Gpc3, Npy1r, Pln,* and *Atp1b2*) that were highly expressed in the pericyte cluster from the single-cell analysis results (*Figure 2C* and *Figure 2—figure supplement 1B*). When we calculate area under the curve (AUC) scores for the markers, *Rgs5* had the highest AUC, but *Rgs5* was also expressed in SMCs in our data (*Figure 2A* and *Figure 2—figure supplement 1C*). *Pln, Ednra, Gpc3,* and *Npy1r* also seemed to be candidate markers, but the literature search excluded these genes as they are also expressed in the SMCs of other tissues or different cell types (*Guo et al., 2020*; *Haghighi et al., 2014*; *Kowalczyk et al., 2015*; *Wittrisch et al., 2020*). *Atp1b2* was also excluded because its protein expression was detected in both mouse cavernous pericytes (MCPs) and aortic SMC using immunofluorescence (IF) staining (*Figure 2—figure supplement 1D*). In the single-cell data, *Lbh* (Limb Bud-Heart) was more specific for pericytes than *Rgs5*, a well-known marker for distinguishing between SMC and pericytes (*Figure 2A and C*). IF staining showed that LBH was optimally expressed in MCP, whereas it was rarely expressed in the SMC of the aorta, consistent with our single-cell data (*Figure 2D* and *Figure 2—figure supplement 1E*). In the blood vessels of other organ such as the bladder, α-SMA-expressing SMC or CD31-expressing endothelial cells were

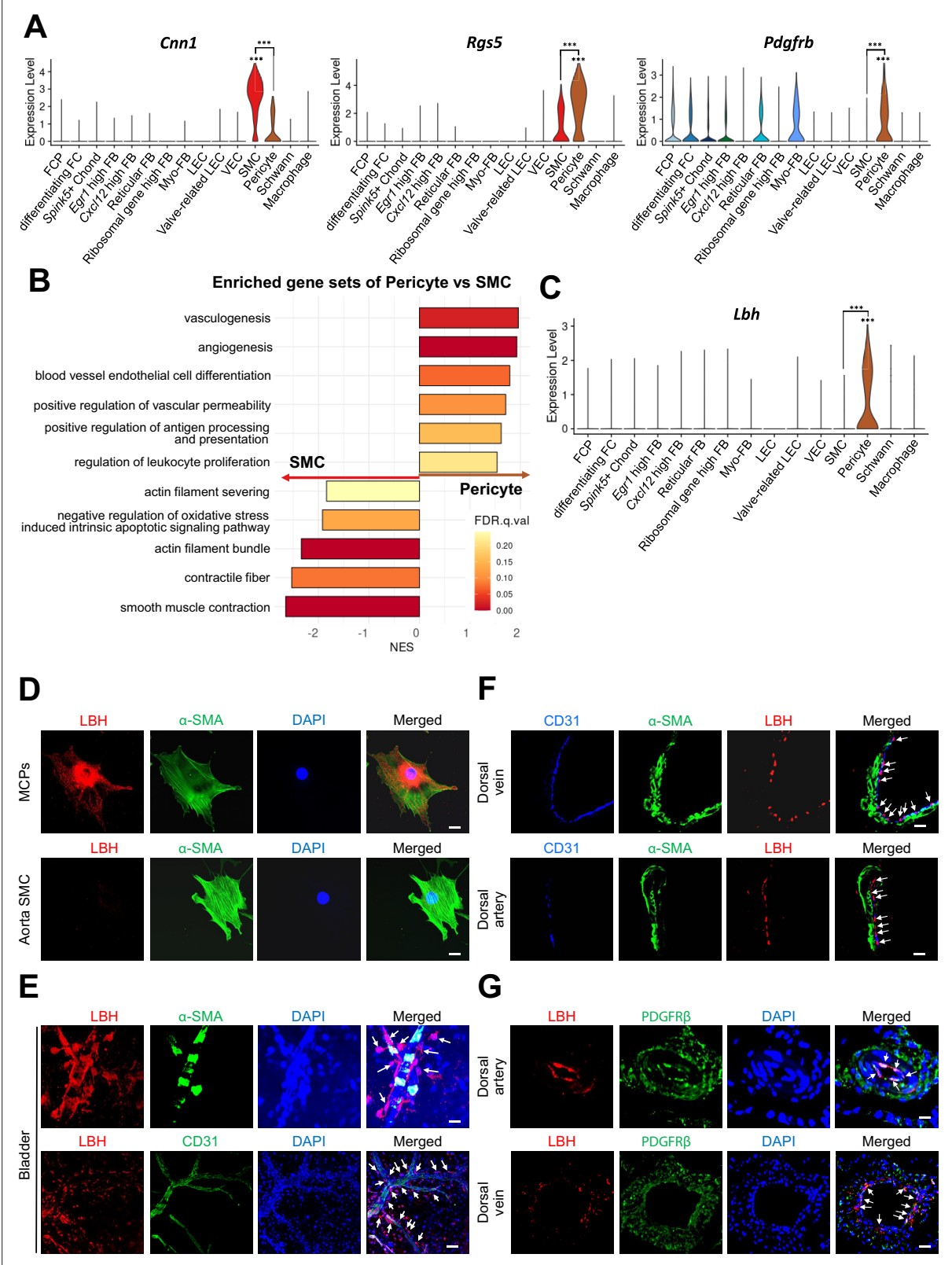

**Figure 2.** Identification of *Lbh* as a marker of pericytes. (**A**) Expression of well-known marker genes of SMC and pericyte (*Cnn1* for SMC, *Rgs5* and *Pdgfrb* for pericyte). (**B**) Significantly enriched gene sets associated with function of SMC and pericyte. Gene sets with positive normalized enrichment score (NES) are enriched in pericyte, and negative values are enriched in SMC. (**C**) Violin plots showing expression of *Lbh* in each cell type. (**D** and **E**) LBH (red)/α-SMA (green) and LBH (red)/CD31 (green) staining in MCPs, aorta SMC, and bladder tissues. Nuclei were labeled with DAPI (blue). Scale

*Figure 2 continued on next page*

Figure 2 continued

bars, 25 µm (MCPs and aorta SMC), 50 µm (bladder top panel) and 100 µm (bladder bottom panel). (**F** and **G**) CD31 (blue)/α-SMA (green)/LBH (red) and LBH (red)/PDGFRβ (green) staining in mouse dorsal vein and dorsal artery tissues. Nuclei were labeled with DAPI (bottom panel, blue). Scale bars, 25 µm (dorsal artery) and 50 µm (dorsal vein). Arrows indicate the LBH expressed pericytes. MCPs, mouse cavernous pericytes; SMC, smooth muscle cell; DAPI, 4,6-diamidino-2-phenylindole. ***p<0.001 by DESeq2 test.

The online version of this article includes the following figure supplement(s) for figure 2:

**Figure supplement 1.** Pericyte-specifically expressed genes in single-cell RNA sequencing data.

**Figure supplement 2.** Expressions of *Lbh* in single-cell RNA sequencing data from tissues other than the penis.

surrounded by Lbh-expressing pericytes but rarely overlap (*Figure 2E*). LBH expression was easily distinguishable from CD31 or α-SMA in mouse dorsal vein and dorsal artery tissues in penile tissue (*Figure 2F*, top panel), and is more specific than traditional pericyte markers (PDGFRβ) (*Figure 2G*, bottom panel). For instance, PDGFRβ is expressed in pericytes and other surrounding tissue cells (as indicated by arrows and dotted area).

Finally, we examined *Lbh* expression patterns in various mouse tissues using the mouse single-cell atlas (Tabula Muris), although endothelial and pericyte clusters were not subclustered in most tissues from Tabula Muris. To identify pericytes, we relied on the expression pattern of known marker genes (*Pecam1* for endothelial cells, *Rgs5*, *Pdgfrb*, and *Cspg4* for pericytes). *Lbh* was expressed in pericytes of the bladder, heart and aorta, kidney, and trachea but not as specifically in penile pericytes (*Figure 2—figure supplement 2A–D*). However, it is worth noting that other known pericyte markers were also did not exhibit exclusive expression in pericytes across all the tissues we analyzed. Therefore, in certain tissues, particularly in mouse penile tissues, *Lbh* may be a valuable marker in conjunction with other established pericyte marker genes for distinguishing pericytes.

## Interactions between pericyte and other cell types in diabetes versus normal

Pericytes regulate various physiological functions involving other cell types such as endothelial cell and SMC. We compared the cell-cell interactions between pericytes and other cell types based on ligand-receptor interactions in single-cell data from mouse cavernous tissues under normal and hyperglycemic conditions. In diabetes, angiogenesis-related interactions, including vascular endothelial growth factor (VEGF) and fibroblast growth factor, were reduced between pericytes and other cell types compared to those under normal conditions (*Figure 3A–C*). In addition, interactions related to neuronal survival, adhesion, migration, and proliferation decreased in diabetic mice, as opposed to an increase in BMP signaling-related interactions (*Figure 3—figure supplement 1* and *Figure 3—figure supplement 2*). In addition to cell-cell interactions, the overall gene sets associated with angiogenesis, adhesion, and cell migration in pericytes were downregulated under diabetic conditions (*Figure 3D*). We performed gene regulatory network analysis to compare the activities of transcription factors (TFs) in pericytes under diabetic and normal conditions. In normal pericytes, TFs promoting angiogenesis (*Klf5*, *Egr1*, *Lmo2*, *Junb*, and *Elk1*) were significantly more active than in diabetic pericytes (*Figure 3E*). In contrast, TFs that inhibit angiogenesis (*Ppard* and *Hoxd10*) were more active in diabetes. Experimental data indicated that the expression level of *Lmo2*, *Junb*, *Elk1*, and *Hoxd10* was higher (*Hoxd10*) or lower (*Lmo2*, *Junb*, *Elk1*) in diabetic pericytes compared to normal pericytes (*Figure 3—figure supplement 3*). Furthermore, using the proteome profiler mouse angiogenesis array, we tested 53 angiogenesis factors in cavernous tissue, among which the expression of OPN (*Dai et al., 2009*), CD105 (*Browne et al., 2018*), TSP-2 (*Kyriakides et al., 2001*), IGFBP2 (*Li et al., 2020*), and IGFBP9 (*Lin et al., 2003*) decreased under diabetic conditions, while MMP-3 (*Abdul et al., 2023*), angiogenin (*Neubauer-Geryk et al., 2012*), and PF4 (*Cella et al., 1986*) increased (*Figure 3F* and *Supplementary file 1*). MMP-3 expression in the neurovascular unit is enhanced under diabetic conditions, leading to vascular damage (*Abdul et al., 2023*). Thus, angiogenesis activity decreased in MCPs under diabetic conditions.

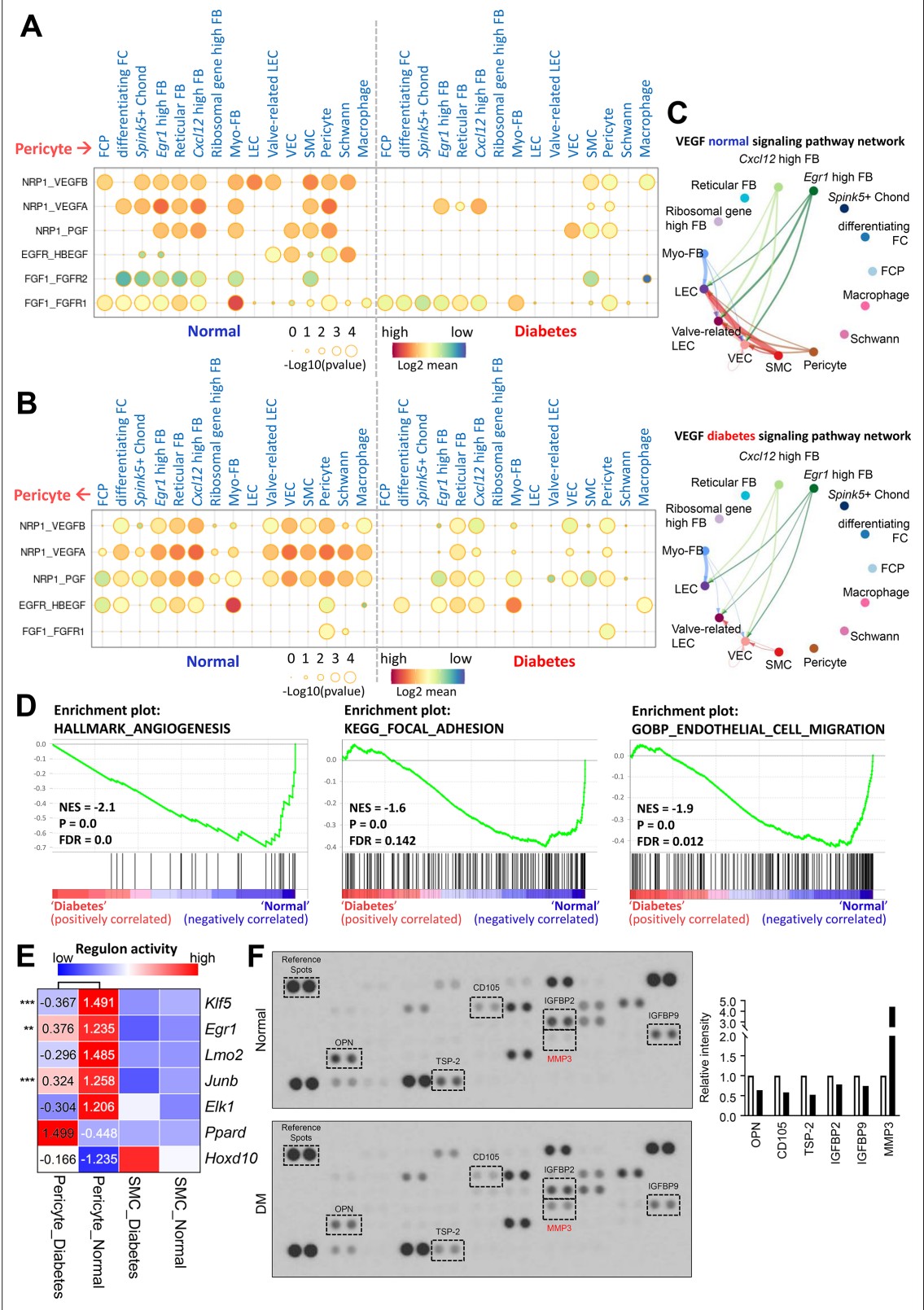

**Figure 3.** Cell-cell interactions between pericytes and other cell types in normal and diabetes. (**A**) CellPhoneDB dot plots showing angiogenesis-associated ligand-receptor interactions from pericytes to other cell types in normal and diabetic samples. p-Values are indicated as circle sizes. The means of the average expression level of the interaction are indicated by color. (**B**) CellPhoneDB dot plots showing angiogenesis-associated ligand-receptor interactions from other cell types to pericytes in normal and diabetic samples. p-Values are indicated as circle sizes. The means of the average

*Figure 3 continued on next page*

*Figure 3 continued*

expression level of the interaction are indicated by color. (**C**) The inferred vascular endothelial growth factor (VEGF) signaling pathway network in normal and diabetes using CellChat. The width of line represents the communication probability. The color of line matches the sender of the signal. (**D**) Gene set enrichment analysis (GSEA) plots showing the gene sets downregulated in diabetic pericytes compared to normal pericytes. (**E**) Heatmap showing the regulon activities of angiogenesis-related transcription factors in pericytes and smooth muscle cells (SMCs) in normal and diabetes. Scaled values of regulon activity scores are displayed on the heatmap. **p<0.01; ***p<0.001 by generalized linear model. (**F**) Proteome profiler mouse angiogenesis array analysis of mouse penis tissues from age-matched control and diabetic mice. The relative expression of each protein was determined by comparing the respective plots to the positive control (reference spot). The frame dot line indicates changed proteins between control and diabetic mice. Expression of the indicated proteins was quantified by assessing the intensity of the dot using ImageJ. DM, diabetes mellitus.

The online version of this article includes the following figure supplement(s) for figure 3:

**Figure supplement 1.** Cell-cell interactions between pericytes and other cell types showing significant differences between normal and diabetes.

**Figure supplement 2.** Cell-cell communication between cell types in normal and diabetes using CellChat.

**Figure supplement 3.** RT-PCR validation of differentially expressed genes from single-cell RNA sequencing analyses in MCPs exposed to NG or HG conditions for 72 hr.

## LBH improves erectile function under diabetic conditions through enhanced neurovascular regeneration

LBH is vital in promoting angiogenesis in human glioma under hypoxic conditions (*Jiang et al., 2019*). We performed IF staining for LBH and another pericyte marker (CD140b) in mouse cavernous tissues to explore the effect of LBH on ED. We found that LBH and CD140b expression significantly decreased in the pericytes under diabetic conditions compared to those in age-matched controls (*Figure 4A and C*). In addition, we assessed the expression of LBH in vitro under diabetic conditions and found that its expression was significantly decreased compared to that under normal conditions in MCPs (*Figure 4B and D*). All IF staining results were confirmed by western blotting (*Figure 4E and F*). In addition, we overexpressed LBH in diabetic mice by intracavernosal injection of lentiviruses containing an ORF mouse clone of *Lbh* and assessed erectile function 2 weeks later (*Figure 4—figure supplement 1*). During electrical stimulation, the ratios of maximal and total intracavernous pressure (ICP) to mean systolic blood pressure (MSBP) were significantly reduced in PBS- and lentivirus ORF control particle-treated diabetic mice compared to age-matched controls. However, LBH overexpression in diabetic mice significantly improved this erection parameter, reaching 84% of the control values (*Figure 4G and H* and *Supplementary file 2*). Moreover, IF staining for CD31 (an endothelial cell marker), NG2 (a pericyte marker), neurofilament-2000 (NF), and neuronal NOS (nNOS) revealed that LBH overexpression significantly improved the endothelial cell, pericyte, and neuronal cell contents in diabetic mice (*Figure 4I–N*). These effects were achieved by inducing the survival cells in the cavernosum tissues (such as decreased apoptosis, increased proliferation, migration, and tube formation) and major pelvic ganglion (MPG) neurite sprouting under diabetic conditions (*Figure 4—figure supplement 2*). Thus, LBH is vital in promoting neurovascular regeneration under diabetic conditions.

## Expression of LBH in human cavernous pericytes and its role in diabetic conditions

Using a previous single-cell transcriptomics data of the human corpus cavernosum (*Zhao et al., 2022*), we examined whether *LBH* is highly expressed in human cavernous pericytes and has a similar role in diabetic conditions (*Figure 5A*). Further clustering of an SMC cluster (marker: *ACTA2*) in the previous study identified three subclusters: ACTA2-expressing cluster as myofibroblasts (MFBs), *CNN1*-expressing cluster as SMCs, and *RGS5*- and *LBH*-expressing clusters as pericytes (*Figure 5B* and *Figure 5—figure supplement 1A and B*). Gene ontology analysis using DEGs between these clusters identified terms related to pericyte function in *LBH*-expressing pericytes (angiogenesis and leukocyte migration) (*Figure 5C*). The newly identified pericyte marker, LBH is also a marker of pericytes in the human corpus cavernosum, and LBH expression was significantly reduced in human cavernous tissues from patients with diabetes-induced ED compared to age-matched controls (*Figure 5D and F*). Furthermore, LBH expression was significantly reduced in primary cultured human cavernous pericytes under diabetic conditions compared to normal glucose (NG) conditions (*Figure 5E and G*).

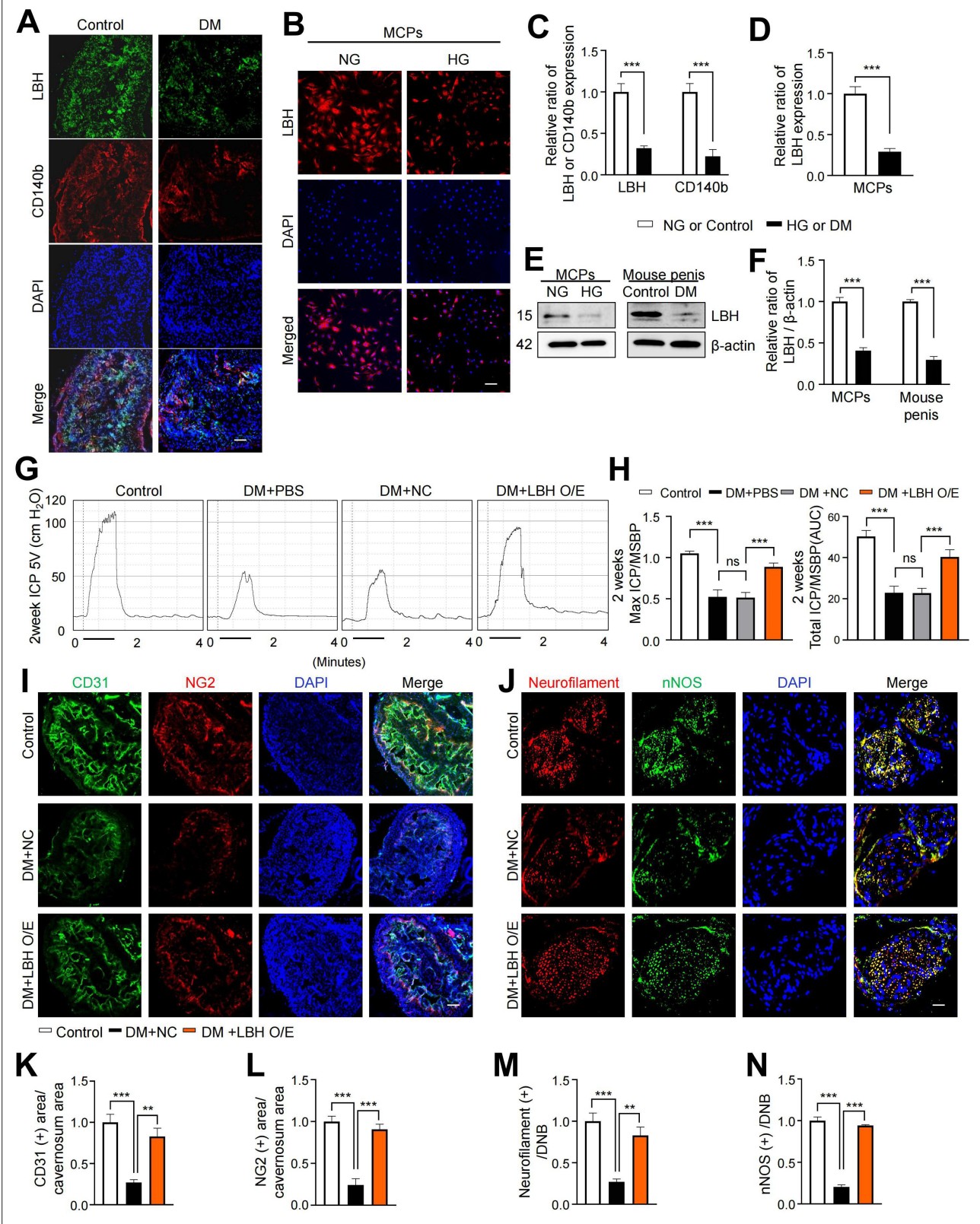

**Figure 4.** LBH improves erectile function under diabetic conditions through induction of neurovascular regeneration. (**A** and **B**) Representative images of immunofluorescence staining of LBH (green)/CD140b (red) in cavernosum tissues and LBH (red) in MCPs under normal and diabetic conditions (in vivo and in vitro). Nuclei were labeled with DAPI (blue). Scale bar, 100 μm. (**C** and **D**) Quantification of LBH or CD140b expression in in vivo and in vitro by using ImageJ, and results are presented as means ± SEM (n=4). (**E**) Representative western blots for LBH of MCPs under NG and HG conditions, and

*Figure 4 continued on next page*

*Figure 4 continued*

mouse penis tissues from age-matched control and diabetic mice. (**F**) Normalized band intensity ratio of LBH to β-actin was quantified using ImageJ, and results are presented as means ± SEM (n=4). (**G**) Representative intracavernous pressure (ICP) responses for the age-matched control and diabetic mice stimulated at 2 weeks after intracavernous injections with PBS, lentiviruses ORF control particles (NC), and ORF clone of mouse *Lbh* (LBH O/E) (20 μL for PBS, 5×10$^4$ IFU/mouse for lentiviral particles). The stimulus interval is indicated by a solid bar. (**H**) Ratios of mean maximal ICP and total ICP (area under the curve) versus MSBP were calculated for each group, and the results are presented as means ± SEM (n=5). (**I** and **J**) Cavernous CD31 (endothelial cell, red), NG2 (pericyte, green), neurofilament (red), and neuronal nitric oxide synthase (nNOS) (green) staining in cavernous tissues from age-matched control (**C**) and diabetic mice stimulated at 2 weeks after intracavernous injections with lentiviruses ORF control particles (NC) and ORF clone of mouse *Lbh* (LBH O/E). Scale bars, 100 μm (left), 25 μm (right). (**K–N**) Quantitative analysis of cavernous endothelial cell, pericyte, and neuronal cell content were quantified by ImageJ, and results are presented as means ± SEM (n=4). The relative ratio in the NG or control group was defined as 1. **p<0.01; ***p<0.001. DM, diabetes mellitus; PBS, phosphate-buffered saline; MCPs, mouse cavernous pericytes; NG, normal glucose; HG, high glucose; DAPI, 4,6-diamidino-2-phenylindole; MSBP, mean systolic blood pressure; ns, not significant.

The online version of this article includes the following figure supplement(s) for figure 4:

**Figure supplement 1.** LBH immunofluorescence staining in corpus cavernosum tissues after infection with lentiviruses containing ORF mouse clone of *Lbh*.

**Figure supplement 2.** LBH enhances pericyte angiogenesis and MPG neurite sprouting under high glucose conditions.

## LBH-interacting protein identification in MCPs

We conducted liquid chromatography tandem mass spectrometry (LC-MS/MS) analysis following the immunoprecipitation of LBH from MCPs to further elucidate the LBH-mediated systematic network in MCPs. In protein-protein interaction (PPI) databases, only CRYAB was experimentally confirmed to interact with LBH (*Pourhaghighi et al., 2020*). However, only the co-fractionation of LBH and CRYAB has been outlined, and direct binding between the two proteins has not been elucidated yet. We identified αB-crystallin (CRYAB) from band 1 and Vimentin (VIM) from bands 2 and 3, co-immunoprecipitated with LBH, VIM, and CRYAB (*Figure 6A and B*). Furthermore, double IF staining indicated that LBH co-localized with CRYAB and VIM in mouse cavernous tissues and MCPs (*Figure 6C and D*). Thus, CRYAB and VIM are novel LBH-interacting proteins in MCPs.

Based on the STRING and BioGrid databases, we reconstructed the PPI network of CRYAB, VIM, and LBH, and the primary and secondary interacting proteins of LBH (*Figure 6E*). To identify the molecular mechanisms underlying interactions of LBH, CRYAB, and VIM, we performed gene ontology analysis using proteins from the PPI network. We found that these proteins are mainly involved in neurogenesis, neuronal development, and the nervous system (*Figure 6F*). In addition, the proteins involved in 'angiogenesis' and 'response to estradiol' that are associated with pericyte activity were also included in this PPI network. Since the expression of LBH was decreased in the pericytes under diabetic conditions, we identified the expression level of angiogenesis and nerve system-related genes in diabetic pericytes using single-cell data. GSEA showed that gene sets related to angiogenesis and the nervous system were enriched in normal pericytes compared to diabetic pericytes (*Figure 6G*). Finally, we found that CRYAB expression significantly reduced in vitro and in vivo under diabetic conditions, whereas VIM expression significantly increased (*Figure 6H and I*). This reveals that the expression of nervous system- and angiogenesis-related genes are downregulated as LBH decreases in pericytes under diabetic conditions, affecting the interacting molecules.

## Discussion

DM is a major cause of ED, and poor long-term glycemic control can lead to nerve and blood vessel damage (*Desai et al., 2023*). Many studies have outlined angiogenic and neurotrophic factors, such as VEGF, Comp-Ang1, dickkopf2, leucine-rich alpha-2 glycoprotein 1, brain-derived neurotrophic factor (BDNF), and neurotrophin-3 that been tested as therapeutic options for ED (*Bennett et al., 2005*; *Burchardt et al., 2005*; *Hu et al., 2018*; *Jin et al., 2011*; *Yin et al., 2018*; *Yin et al., 2022*). However, poor efficacy, side effects such as inflammation, and complex drug-protein engineering have limited their success in clinical trials. A more detailed understanding of the intercellular signaling mechanisms and microenvironment in the penis under physiological and pathological conditions are necessary to provide more effective therapeutic targets. We therefore employed single-cell RNA sequencing to dissect the complex transcriptional changes in various cell types in ED using a diabetes-induced ED mouse model, as diabetic ED accounts for 75% of ED patients (*Sáenz de Tejada et al., 2002*).

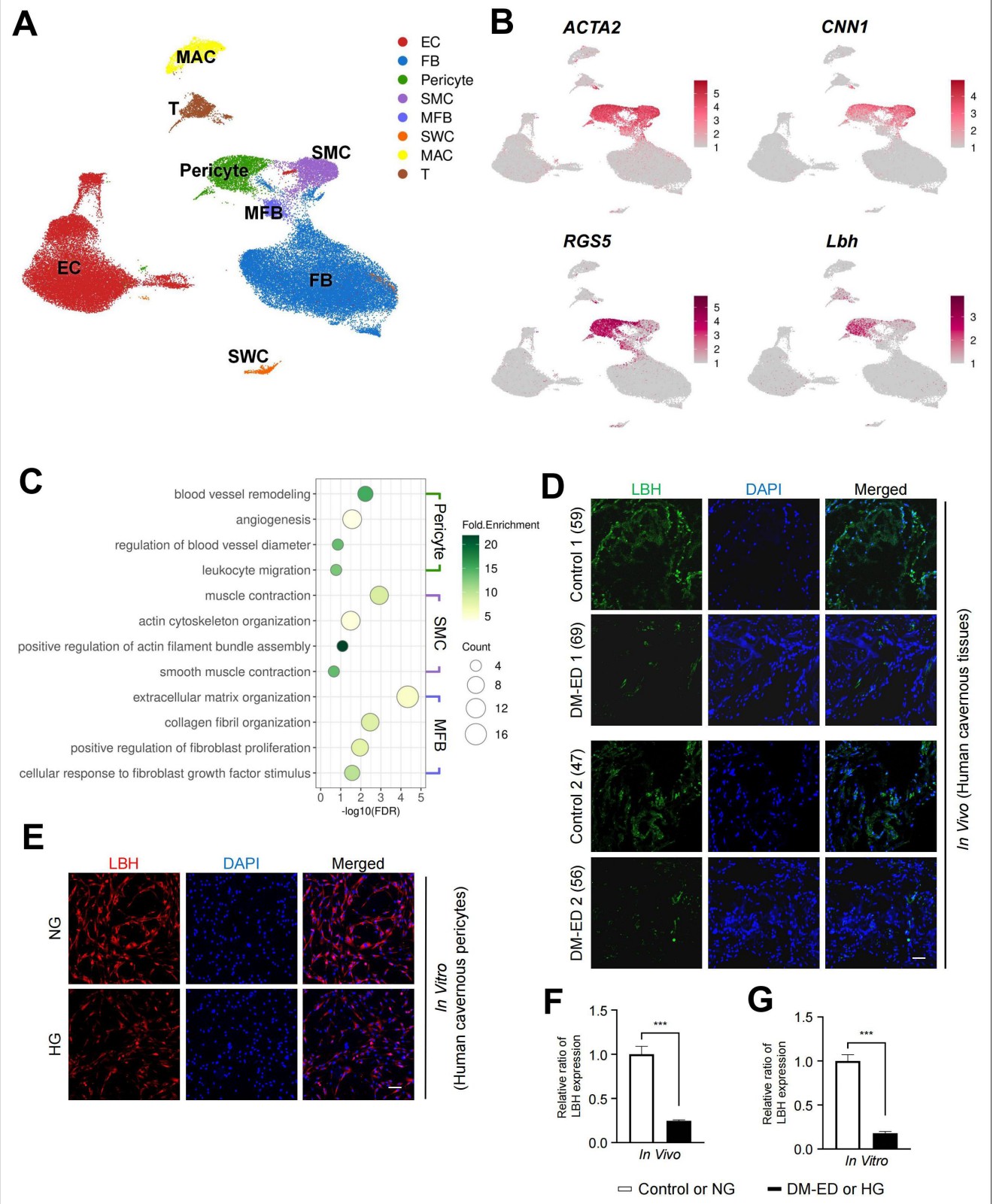

**Figure 5.** LBH as a marker of pericyte in human corpus cavernosum. (**A**) Visualization of single-cell data from human corpus cavernosum using Uniform Manifold Approximation and Projection (UMAP). Each cell type is indicated by a different color. EC, endothelial cells; FBs, fibroblasts; SMC, smooth muscle cell; MFB, myofibroblast; SWC, Schwann cell; MAC, macrophage; T, T cell. (**B**) Expression of marker genes of SMC (*ACTA2* and *CNN1*) and marker genes of pericyte (*RGS5* and *LBH*). (**C**) Biological processes identified through gene ontology analysis of clusters annotated as pericyte, SMC,

*Figure 5 continued on next page*

*Figure 5 continued*

and MFB. (**D**) LBH (green) staining in cavernous tissues from two patients with diabetic erectile dysfunction and two patients with congenital penile curvature who had normal erectile function during reconstructive penile surgery. Scale bar, 100 µm. (**E**) LBH (red) staining in primary cultured human cavernous pericytes under NG and HG conditions for 3 days. (**F** and **G**) LBH immunopositive areas were quantified by ImageJ, and results are presented as means ± SEM (n=4). Nuclei were labeled with DAPI (blue). The relative ratio in the control or NG group was defined as 1. ***p<0.001. NG, normal glucose; HG, high glucose; DM, diabetes mellitus; DAPI, 4,6-diamidino-2-phenylindole.

The online version of this article includes the following figure supplement(s) for figure 5:

**Figure supplement 1.** Expression of marker genes of pericyte and smooth muscle cells in human penis single-cell RNA sequencing data.

Our findings from in vivo animal models and in vitro experiments were confirmed using human data and samples, enabling us to understand how the results from mouse models could be translated to humans and developed as novel treatment targets.

It is well known that ED is a complex neurovascular disease (*Leung et al., 2004*), but the mechanism by which diabetes causes ED is still unclear. Many studies have shown that type 1 diabetes induces ED through different physiological processes, such as affecting penile contraction and relaxation, immune inflammatory response, neuropathy, structural and functional disturbance of cavernous SMCs, and dysfunction of endothelial cells, FBs, and pericytes (*Liu et al., 2023*; *Thorve et al., 2011*; *Yin et al., 2015*). Although other major cell populations in penile tissue such as SMCs, endothelial cell, and FBs have been extensively studied, pericytes have mainly been investigated in the context of the central nervous system (CNS). For example, in the CNS, pericytes are involved in maintaining the integrity of the brain's blood-brain barrier (*Ferland-McCollough et al., 2017*), regulating blood flow at capillary junctions (*Gonzales et al., 2020*), and promoting neuroinflammatory processes (*Brown et al., 2019*), whose dysfunction is considered an important factor in the progression of vascular diseases such as Alzheimer's disease (*Winkler et al., 2014*). But little is known about the role of pericytes in penile tissue (*De Leonardis et al., 2022*; *de Souza et al., 2022*; *Yin et al., 2015*). In order to explore the role of pericytes in repairing the corpus cavernosum vascular and neural tissues damaged by DM, we focused on pericytes, which are multipotent perivascular cells that contribute to the generation and repair of various tissues in response to injury (*Birbrair et al., 2015*). Although recent studies have shown that pericytes are involved in physiological mechanisms of erection, little is known about their detailed mechanisms. Many known pericyte markers are co-expressed in other cell types; thus, identifying the specific marker genes for pericytes in vivo is the first step characterizing pericyte cell type. From the pericyte cluster in the single-cell RNA sequencing results, we found that *Lbh, Ednra, Gpc3, Npy1r, Pln,* and *Atp1b2* are specific pericyte markers (*Figure 2—figure supplement 1B*). LBH was identified as a pericyte-positive and SMC-negative marker that was also highly expressed in human cavernous pericytes (*Figure 5*) and was validated through IF staining in many vascular tissues and cell types (*Figure 2*). As it is widely known, pericytes are primarily located in capillaries, where they surround endothelial cells of blood vessels (*Avolio and Madeddu, 2016*). In our recent studies, we observed minimal overlap in staining between LBH and α-SMA and between LBH and CD31 in penile tissue (*Figure 2* and *Figure 4—figure supplement 1A*), suggesting that the cells expressing LBH were not SMCs or endothelial cells but possibly pericyte-like cells. In small vessels within the bladder, we also found that LBH-expressing cells surrounding CD31-expressing vessels had minimal overlap with α-SMA-expressing SMCs, consistent with the known characteristics of pericytes. In addition, in the triple staining of CD31, LBH, and α-SMA, we clearly found that LBH-expressing cells in the dorsal veins and dorsal artery showed LBH+/α-SMA-/CD31- staining (shown in *Figure 2F*). These results also indicate that the LBH-expressing cells and endothelial cell layer are completely distinct from the SMC layer. LBH-expressing cells are mostly distributed next to and outside endothelial cells. The distribution of LBH-expressing cells was similar to that of known pericytes. Further research is needed to comprehend the differences in LBH expression and its characteristics in other vascular tissues.

Lbh, a highly conserved transcription cofactor, is known to participate in early limb and heart development (*Briegel and Joyner, 2001*). Moreover, LBH can directly target the Wnt signaling pathway (*Rieger et al., 2010*) and regulate neural crest cell development (*Powder et al., 2014*). In addition, *Jiang et al., 2019* showed that LBH overexpression promotes angiogenesis via VEGF-mediated ERK signaling. In contrast, LBH inhibits cell migration and angiogenesis in nasopharyngeal carcinoma (*wu et al., 2022*; *Wu et al., 2021*). These inconsistent results may be related to different histopathological environments. Thus, LBH has dual effects on the development of blood vessels and the nervous

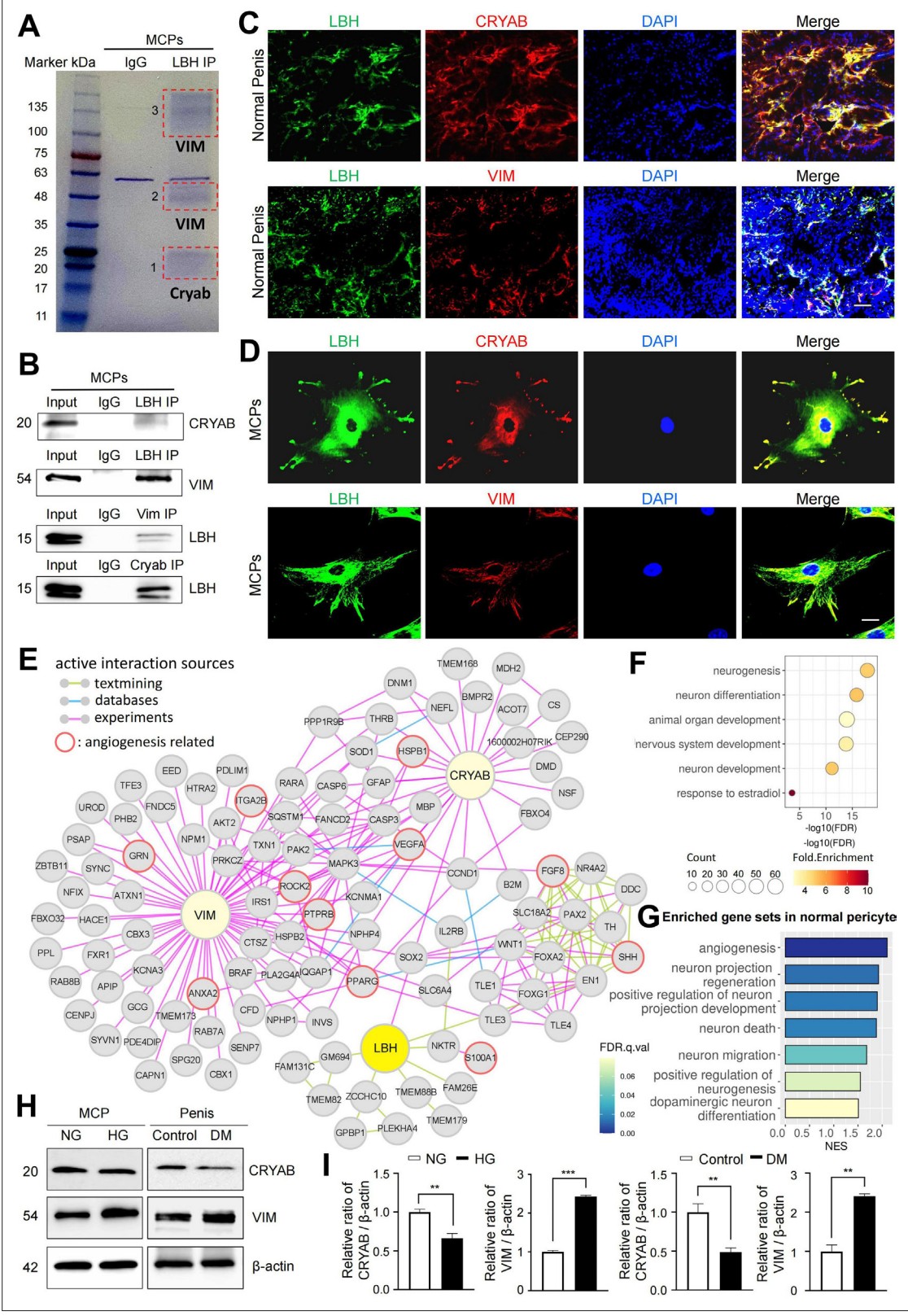

**Figure 6.** LBH-interacting protein identification in mouse cavernous pericytes. (**A**) LBH was immunoprecipitated (IP) from whole-MCPs lysates, resolved on SDS-PAGE gels, and stained with Coomassie blue solution. Gel bands indicated by red frame dot line were analyzed by liquid chromatography tandem mass spectrometry (LC-MS/MS) analysis. (**B**) Co-IP of LBH, CRYAB, and Vimentin from whole-MCPs lysates followed by immunoblot analysis to detect CRYAB, Vimentin, and LBH. (**C** and **D**) Representative images of immunofluorescence staining of LBH (green)/CRYAB (red) and LBH (green)/VIM

*Figure 6 continued on next page*

*Figure 6 continued*

(red) in normal penis tissues and MCPs. Nuclei were labeled with DAPI (blue). Scale bar, 100 μm (top), 25 μm (bottom). (**E**) Protein-protein interaction (PPI) network of LBH, CRYAB, VIM, and first and second interactors of LBH. Lines connecting molecules show interaction sources in color. (**F**) Biological pathways involving molecules in PPI network identified by gene ontology analysis. (**G**) Significantly enriched gene sets associated with angiogenesis and nerve system in normal pericytes compared to diabetic pericytes in single-cell data. (**H**) Representative western blots for CRYAB and VIM of MCPs under NG and HG conditions (left), and mouse penis tissues from age-matched control and diabetic mice (right). (**I**) Normalized band intensity ratio of CRYAB and VIM to β-actin was quantified using ImageJ, and results are presented as means ± SEM (n=4). The relative ratio in the NG or control group was defined as 1. **p<0.01; ***p<0.001. MCPs, mouse cavernous pericytes; NG, normal glucose; HG, high glucose; DM, diabetes mellitus; DAPI, 4,6-diamidino-2-phenylindole.

system, similar to other angioneurins such as VEGF and BDNF (*Zacchigna et al., 2008*). Here, we explored the role of Lbh in the penile tissues of diabetic mice. Interestingly, Lbh expression was significantly lower in cavernous pericytes in a diabetic mouse model. Therefore, we hypothesized that the restoration of Lbh expression in the penis of diabetic mice will improve erectile function. To test our hypothesis, we utilized the diabetes-induced ED mouse model, commonly employed in various studies focusing on microvascular complications associated with type 1 diabetes. We observed that the overexpression of LBH in diabetic mice led to the restoration of reduced erectile function by enhancing neurovascular regeneration. However, this study primarily demonstrated the observed phenomenon without delving into the detailed mechanisms. In addition, our findings suggest that LBH can affect the function of cavernous pericytes, although we cannot definitively conclude which specific cavernous cell types are affected by the overexpressed LBH, whether it be cavernous endothelial cells, SMCs, or others. Subsequent research will be required to conduct more comprehensive mechanistic investigations, such as in vitro studies using cavernous endothelial cells, SMCs, and FBs to address these knowledge gaps. In summary, our results support the hypothesis that LBH can restore diabetes-induced ED by facilitating neurovascular regeneration (*Figure 3*).

Previous studies have shown that CRYAB promotes angiogenesis (*Dimberg et al., 2008*) and interacts with LBH in nasopharyngeal carcinoma cells, where it enhances cell survival by inhibiting the autoproteolytic maturation of caspase-3 (*Wu et al., 2021*). Interestingly, our LC-MS/MS analysis of LBH following immunoprecipitation of MCPs identified VIM as a novel interaction partner and CRYAB (*Figure 5*). In addition, we assessed the expression of CRYAB and VIM under diabetic conditions. Unexpectedly, the expression of VIM was increased, whereas that of CRYAB was downregulated in the penis of diabetic mice and MCPs under diabetic conditions. VIM is a type III intermediate filament protein expressed in mesenchymal cells (*Dave and Bayless, 2014*; *Guo et al., 2013*). It is involved in cell adhesion, migration, cellular integrity, epithelial-mesenchymal transition, and the malignant transformation and metastatic spread of cancer cells (*Chen et al., 2021*; *Dave and Bayless, 2014*; *Liu et al., 2015*). It is a multifunctional protein that interacts with several other proteins with many functions under various pathophysiological conditions (*Danielsson et al., 2018*). For example, VIM interacts with the insulin-like growth factor 1 receptor, promotes axonal extension, and serves as a double-edged sword in the nervous system by regulating axonal regeneration, myelination, apoptosis, and neuroinflammation (*Chen et al., 2023*). Furthermore, *VIM* knockout mice were less susceptible to bacterial infections and had a reduced inflammatory response than wild-type mice (*Moisan et al., 2007*). Recently, *VIM* deficiency was shown to prevent obesity and insulin resistance in type 2 DM by reducing CD36 expression on plasma membranes and intracellular trafficking of glucose transporter type 4 in adipocytes (*Luo et al., 2021*). Therefore, the combination of LBH with CRYAB and VIM could improve neurovascular regeneration in diabetic ED by activating the angiogenic effects of CRYAB and potentially reducing the inflammatory effects of VIM. Collectively, the discovery of the LBH-CRYAB-VIM interaction opens the door to a more specific understanding of the role of pericytes in diabetic ED. However, the clinical implications of these results remain uncertain. Future research endeavors should focus on incorporating LBH-CRYAB-VIM into therapeutic strategies for diabetes and ED.

As far as our knowledge extends, this study marks the first demonstration of LBH's efficacy in restoring erectile function in mice with diabetes-induced ED. This not only introduces a novel approach to addressing diabetic ED but also carries substantial clinical relevance in mitigating other complications associated with diabetes.

## Methods

### Ethics statement and animal study design

Eight-week-old male C57BL/6 mice (Orient Bio, Seongnam-si, Gyeonggi-do, Korea) were used in this study. Animal experiments were approved by the Institutional Animal Care and Use Subcommittee of Inha university (approval number: 200910-719). Animals were monitored daily for health and behavior as previous studies described (*Ghatak et al., 2022*; *Yin, 2022*). Briefly, mice were maintained at room temperature (RT) (23 ± 2°C), 40–60% relative humidity, 12 hr light/dark cycle, and specific pathogen-free conditions. Sixty adult male C57BL/6 mice were used for single-cell sequencing analysis, MCP culture (in vitro study), and erectile function evaluation (in vivo study). All animals were anesthetized with intramuscular injections of ketamine (100 mg/kg, Yuhan Corp., Seoul, Korea) and xylazine (5 mg/kg, Bayer Korea, Seoul, Korea). Animals were euthanized by 100% $CO_2$ gas exchange in a closed container at a $CO_2$ exchange rate of 10–30% container volume/min. In this study, a total of 90 eight-week-old male C57BL/6 mice were used (10 for single-cell RNA sequencing; 40 for MCP culture and related experiments; 20 for erectile function evaluated experiment; 20 for angiogenesis array and other fluorescence examinations). DM was induced as described previously (*Jin et al., 2009*). Shortly, low doses of streptozotocin (STZ, 50 mg/kg, i.p., Sigma-Aldrich, St. Louis, MO, USA) was injected for 5 consecutive days. Eight weeks later, only mice with a tail vein blood glucose level higher than 300 mg/dL and significantly decreased body weights were considered to have DM. Fasting and postprandial blood glucose levels were determined using an Accu-Check blood glucose meter (Roche Diagnostics, Mannheim, Germany) before starting all in vivo studies. Body weights of STZ-treated (diabetic) mice were significantly lower than those of age-matched non-diabetic controls. Fasting and postprandial blood glucose concentrations were significantly higher in diabetic mice than in non-diabetic controls. However, diabetic mice infected with lentiviruses ORF control particles or lentiviruses containing ORF mouse clone of *Lbh* had no body weights and blood glucose levels change. MSBP were similar in three STZ-treated groups and the non-diabetic control group (*Supplementary file 2*). No mice died during erectile function evaluation experiment, and all experiments were performed in a blinded manner.

To test the efficacy of LBH on erectile function, the mice were distributed into four groups as follows: control non-diabetic mice (n=5) and mice with STZ-induced diabetes receiving one successive intracavernous injections of phosphate-buffered saline (n=5, PBS, 20 µL), lentiviruses ORF control particles (n=5, NC, 5×10⁴ infection units in 20 µL; Origene Technology, Rockville, MD, USA) or lentiviruses containing ORF mouse clone of *Lbh* (n=5, LBH O/E, 5×10⁴ infection units in 20 µL; Origene Technology) into the midportion of the corpus cavernosum. A vascular clamp was used to pressurize the bottom of penises immediately before injection and was left in place for 30 min to restrict blood outflow.

### Single-cell RNA sequencing

After mice were euthanized, the penis tissues (n=5 for each group) were harvested, minced, and digested using Multi Tissue dissociation kit (Miltenyi, 130-110-201) with minor modifications as described previously (*Park et al., 2018*). Due to the small size of mouse penile tissues, we pooled five corpus cavernosum tissues for each group. Briefly, tissues were homogenized using 21G and 26 1/2G syringes. The tissues were digested with 50 µL of Enzyme D, 25 µL of Enzyme R, and 6.75 µL of Enzyme A in 500 µL of Dulbecco's modified Eagle medium (DMEM) and incubated for 10 min at 37°C. The enzymes were deactivated by 10% FBS and solution was then passed through tip strainer (70 µm and then 40 µm). After centrifugation at 1000 RPM for 5 min, cell pellet was incubated with 1 mL of RBC lysis buffer on ice for 3 min. After the cell number and viability were analyzed by using Countess AutoCounter (Invitrogen, C10227), the single-cell suspension was loaded onto 10× Chromium Single Cell instrument (10× Genomics). Barcoding and cDNA synthesis were performed according to the manufacturer's instructions.

### Quality control, clustering, and cell-type annotation

FASTQ files from sequencing were aligned to the mouse reference sequence (mm10) using CellRanger count pipeline (10× Genomics, v3.0.2). Filtered feature-barcode matrices outputs by CellRanger were used to create Seurat objects for single-cell RNA sequencing data analysis (Seurat v3.1.5). Cells with the number of genes less than 200 were filtered out from Seurat objects. Quality controlled data were normalized using NormalizeData function with default parameters. Highly variable features

were identified using the FindVariableFeatures function with default parameters. Data were scaled by setting the number of UMI as variables to regress out using ScaleData function. The scaled data were merged into one Seurat object, and then normalized, and highly variable genes were identified and scaled. Linear dimensional reduction performed using RunPCA function with 30 principal components. The RunHarmony function was used to correct the batch effect between the merged data. Cells were clustered based on similar feature expression patterns using FindClusters function (resolution = 1.5) and FindNeighbors (reduction = 'harmony', dims = 1:20). To visualize and explore data, non-linear dimensional reduction was performed using Uniform Manifold Approximation and Projection. To merge clusters with similar gene expression patterns, FindMarkers function (test.use = 'bimod') parameters were used to find DEGs, and cluster pairs with less than 10 DEGs (adjusted p-value<0.01, log_fold change≥1) were merged. DEGs of each cluster were found using FindAllMarkers function ( max.cells.per.ident=100, min.diff.pct=0.3, only.pos=TRUE).

The clusters were annotated by expression of marker genes (*Comp* and *Fn1* for Chonds; *Col1a1*, *Col1a2,* and *Fbln1* for FBs; *Mylk* and *Col3a1* for fibrochondrocytes progenitor; *Prg4* and *Anxa8* for differentiating fibrochondrocyte; *Mgp* and *Mfap5* for reticular FBs; *C7*, *Lum*, and *Mmp2* for MFB; *Prox1*, *Ccl21a*, and *Lyve1* for LEC; *Foxc2* for valve-related LEC; *Eng* and *Flt1* for VEC; *Cnn1,Acta2, Myh11*, *Tagln*, and *Actg2* for SMC; *Rgs5,Cspg4*, and *Pdgfrb* for pericyte; *Cadm4* and *Plp1* for Schwann cell; *Cd68*, *C1qa,* and *C1qb* for macrophage). The expressions of these marker genes were identified by FeaturePlot, VlnPlot, and FindAllMarkers function from Seurat package.

## Identification of DEGs and gene ontology analysis

DEGs between two groups were identified using FindMarkers function (test.use = 'MAST') from Seurat R package. Among the DEGs found by FindMarkers based on Bonferroni correction, only genes with |log_fold change|>0.25 and adjusted p-value<0.05 were considered as significant DEGs. DEGs were visualized as volcano plots. AUC scores of *Rgs5*, *Pln*, *Ednra*, *Npylr*, *Atp1b2*, and *Gpc3* for ability of a binary classifier to distinguish between pericyte and the other cell types in mouse penile tissues were measured by using FindMarkers function (test.use = 'roc'). A pseudobulk DEG analysis for Lbh vs SMC or the other cell types was performed by using FindMarkers function (test.use = 'DESeq2'). For the DESeq2 test, we added a pseudocount of 1 to all read counts. Gene ontology analysis was performed using DAVID with significant DEGs as inputs. Gene ontology analysis on molecules of the PPI network included parent terms (GOTERM_BP_5), and the others were gene ontology mappings directly annotated by the source database (GOTERM_BP_DIRECT). Only terms with p-value<0.05 and false discovery rate (FDR)<0.25 were considered significant gene ontology.

## Gene set enrichment analysis

Enriched gene sets between two groups were identified using GSEA (http://www.broadinstitute.org/gsea/index.jsp) with 1000 gene set permutation. Curated (c2), ontology (c5) gene sets, and hallmarks (h) gene sets were selected as gene sets database. A 'Signal2Noise' metric was used for ranking genes, and gene sets larger than 500 or smaller than 5 were excluded from the analysis. Only gene sets with p-value<0.05 and FDR<0.25 were considered significant enriched gene sets.

## Single-cell data of Tabula Muris

To determine whether *Lbh* could also be a marker of pericyte in other tissues, we analyzed single-cell RNA data from Tabula Muris (ref: https://doi.org/10.1038/s41586-018-0590-4). We downloaded and analyzed cell-type annotated Seurat R objects. We used data from the microfluidic droplet method only. We subclustered pericytes from clusters annotated as endothelial cells by identifying the expression of known marker genes (*Pecam1* for endothelial cells, *Rgs5*, *Pdgfrb*, *Cspg4* for pericytes, *Rgs4* for heart pericytes, *Kcna5* for kidney pericytes). The expression of known marker genes and *Lbh* was confirmed through Seurat package's FeaturePlot and VlnPlot function.

## Single-cell gene regulatory network analysis

TF activity of normal or diabetic pericyte were identified using SCENIC R package (v1.1.2-01). To avoid narrow comparison of TF activity in pericytes between normal and diabetic conditions that could lead to faulty reasoning, we included SMCs in the comparison as well. Raw count matrices of SMC and pericyte extracted from the Seurat object were used as input for SCENIC. 10 kb around the transcription

start site (TSS) and 500 bp upstream of the TSS were selected for motif ranking. Genes in input matrix were filtered using geneFiltering function with default parameters. To split targets into positive and negative correlated targets, the correlation was calculated using runCorrelation function. Potential TF targets were inferred using runGenie3 function. Building and scoring the gene regulatory network were performed using runSCENIC functions. Clustering and dimensionality reduction on the regulon activity were performed using tsneAUC function. Only angiogenesis-related TFs with differences in activity between normal and diabetic pericytes were visualized in the heatmap. To assess differential regulon activities of TFs between diabetic pericytes and normal pericytes, we used generalized linear model using the scaled activity scores for each cell from SCENIC as input. We ran the generalized linear model using glm function in R programming.

## Single-cell data of human corpus cavernosum analysis

To determine whether *LBH* could also be a marker of pericyte in human, we analyzed single-cell RNA data of human corpus cavernosum from previous study (*Zhao et al., 2022*). Quality control was performed according to the method of previous study. Quality controlled data were processed according to our data analysis methods. Cells were clustered using FindClusters function (resolution = 0.5). The DEGs of three clusters expressing the SMC markers *ACTA2*, *MYH11*, and *ACTG2*, used in the previous study, were found with the FindAllMarker function. We defined significant DEGs as genes with log_fold change>0.25, adjusted p-value<0.05, and non-overlapping DEGs in different clusters. These significant DEGs were used for gene ontology analysis for cell-type identification.

## Cell-cell interaction analysis

Ligand-receptor communications between cell types were predicted using CellPhoneDB (https://github.com/Teichlab/cellphonedb; *Teichlab, 2021*) and CellChat (https://github.com/sqjin/CellChat; *Jin, 2023*). Normalized single-cell data was used as input for analysis. We performed CellPhoneDB with the statistical method. Interactions that did not differ by more than 0.25-fold mean values between diabetes and normal were considered to have no significant difference. Even if there is a difference in the mean value between diabetes and normal, interactions with p-value of 1 in both diabetes and normal were considered insignificant. The outputs of CellPhoneDB were visualized using ktplot in R. CellChat was performed according to tutorial for comparison analysis of multiple datasets with different cell-type compositions. CellPhoneDB permutes the cluster labels of all cells 1000 times and calculates the mean (mean [molecule 1 in cluster X], mean [molecule 2 in cluster Y]) at each time for each interaction pair, for each pairwise comparison between two cell types. We only considered interactions in which the difference in means calculated by these permutations were greater than 0.25-fold between diabetes and normal. Also, we considered that there is no interaction when the p-value is 1 and that the interactions with p-value<0.05 were significant.

## PPI network visualization

PPIs of LBH, CRYAB, VIM, and first and second interactors of LBH were found by BioGRID and STRING database (organisms: *Mus musculus*). In the STRING database, textmining, experiments, and databases were used as active interaction sources, and only interactions with a minimum required interaction score higher than 0.7 were identified. The identified PPIs were visualized using Cytoscape (*Shannon et al., 2003*).

## Cell culture

Mouse aortic SMCs (Aorta SMC; CRL-2797, ATCC) and human pericytes from placenta (hPC-PL; C-12980, PromoCell) were authenticated according to ATCC and PromoCell guidelines and used within 6 months of receipt. Aorta SMC and hPC-PL were cultured in complement DMEM (Gibco, Carlsbad, CA, USA) supplemented with 10% FBS, and 1% penicillin/streptomycin (Gibco), and incubated the cells at 37°C in a 5% $CO_2$ atmosphere.

For the primary culture of MCPs, follow the protocol described previously (*Neng et al., 2013*; *Yin et al., 2020*), in brief, the cavernous tissue was cut into several pieces around 1 mm, and the pieces settled by gravity to the collagen I-coated 35 mm cell culture dishes (BD Biosciences). After 30 min incubation at 37°C with 300 μL complement of DMEM, 10% FBS, 1% penicillin/streptomycin, and 10 nM human pigment epithelium-derived factor (Sigma-Aldrich), we added an additional 900 μL

complement medium and incubated the samples at 37°C in a 5% $CO_2$ atmosphere. Change the medium every 2 days. After the cells are confluent and spread to the bottom of the dish (approximately 2 weeks after the start of culture), subculture using only sprouting cells. The sprouting cells were seeded onto dishes coated with 50 µL/ml collagen I (Advanced BioMatrix). Cells between passages 2 and 4 were used for experiments. In order to examine the effect of LBH overexpression under NG (5 mM glucose, Sigma-Aldrich) or high glucose (HG, 30 mM glucose, Sigma-Aldrich) conditions, MCPs were infected with lentiviruses ORF control particles (NC, $5\times10^5$ infection units per milliliter cultured medium; Origene Technology) or ORF clone of mouse *Lbh* (LBH O/E, $5\times10^5$ infection units per milliliter cultured medium; Origene Technology) under HG conditions for at least 3 days.

## Human corpus cavernosum tissue and cavernous pericyte culture

For fluorescence examinations, human corpus cavernosum tissues were obtained from two patients with congenital penile curvature (59 and 47 years of age) who had normal erectile function during reconstructive penile surgery and two patients with diabetic ED (69 and 56 years of age) during penile prosthesis implantation. For primary pericyte culture, the fresh adult corpus cavernosum tissues from patients (59 years of age) with congenital penile curvature who have normal erectile function during reconstructive penile surgery were collected after surgery and transferred into sterile vials containing Hank's balanced salt solution (Gibco, Carlsbad, CA, USA) and washed twice in PBS. The detailed methods of the human cavernous pericytes were isolated and maintained as described previously (*Neng et al., 2013*; *Yin et al., 2020*). Human cavernous pericytes at passages 2 or 3 were used for experiments. All tissue donors provided informed consent, and the experiments were approved by the internal review board of our university.

## RT-PCR

Total RNA was extracted from cultured cells using Trizol (Thermo Fisher Scientific) following the manufacturer's protocols. Reverse transcription was performed using 1 µg of RNA in 20 µL of reaction buffer with oligo dT primer and AccuPower RT Premix (Bioneer Inc, Daejeon, Korea). We used the following primers to measure relative changes in mRNA levels: mouse *Lmo2* forward, 5'-atccctgctgac atgtggtg-3'; mouse *Lmo2* reverse, 5'-cccagcttgtagtagaggcg-3'; mouse *Junb* forward, 5'-aggcagctactt ttcgggtc-3'; mouse *Junb* reverse, 5'-ccagggctttgacaaaaccg-3'; mouse *Elk1* forward, 5'-gtagggatcaaa cggtcacctt-3'; mouse *Elk1* reverse, 5'-taaagacgtgtgcctctaccac-3'; mouse *Hoxd10* forward, 5'-agagtggc agaaagaagaggtg-3'; mouse *Hoxd10* reverse, 5'-cttgagtttcattcggcggtt-3'; *GAPDH* forward, 5'-ccac tggcgtcttcaccac-3'; *GAPDH* reverse, 5'-cctgcttcaccaccttcttg-3'. The PCR was performed with denaturation at 94°C for 30 s, annealing at 60°C for 30 s, and extension at 72°C for 1 min in a DNA Engine Tetrad Peltier Thermal Cycler (MJ Research). For the analysis of PCR products, 10 µL of each PCR was electrophoresed on 1% agarose gel and detected under ultraviolet light. GAPDH was used as an internal control.

## Measurement of erectile function

Erectile function was measured after mice were anesthetized by intraperitoneal injection of ketamine (100 mg/kg) and xylazine (5 mg/kg) as described previously (*Jin et al., 2009*). Briefly, the cavernous nerve was stimulated for 1 min using bipolar platinum wire electrodes at 1 or 5 V at 12 Hz and a pulse width of 1 ms condition. Each stimulation was repeated at least two times every 10 min. Record the maximal ICP and total ICP during stimulation. The AUC from the initiation of cavernous nerve stimulation to 20 s after stimulation termination was measured as total ICP. Before ICP measurements, the systemic blood pressure was measured continuously using a noninvasive tail-cuff system (Visitech Systems, Apex, NC, USA). The ratio of maximal ICP and total ICP to MSBP was calculated to normalize variations in systemic blood pressure.

## TUNEL assay

The MCPs cell death after infected with lentiviruses ORF control particles (NC, $5\times10^4$ infection units per milliliter cultured medium; Origene Technology) or ORF clone of mouse *Lbh* (LBH O/E, $5\times10^4$ infection units per milliliter cultured medium; Origene Technology) under NG or HG conditions were evaluated by TUNEL (terminal deoxynucleotidyl transferase-mediated deoxyuridine triphosphate nick-end labeling) assay using the ApopTag Fluorescein In Situ Apoptosis Detection Kit (S7160,

Chemicon, Temecula, CA, USA) according to the manufacturer's instructions. Numbers of TUNEL-positive apoptotic cells were obtained by a confocal fluorescence microscope.

## Cell migration assay

MCPs migration assays were performed using the SPLScar Block system (SPL Life Sciences, Pocheon-si, Gyeonggi-do, Korea) on 60 mm culture dishes. In brief, MCPs were infected with lentiviruses ORF control particles (NC, $5 \times 10^4$ infection units per milliliter cultured medium; Origene Technology) or ORF clone of mouse *Lbh* (LBH O/E, $5 \times 10^4$ infection units per milliliter cultured medium; Origene Technology) under NG or HG conditions. Then conditioned cells were seeded into three-well blocks at >90% confluence. Blocks were removed after 5 hr and cells were incubated for additional 24 hr in DMEM containing 2% FBS and thymidine (2 mM, Sigma-Aldrich). The images were taken using a phase-contrast microscope (Olympus), and cell migration was analyzed by determining the percentage of cells that moved into the frame line shown in the figures from four separate block systems in a blinded manner using ImageJ software (National Institutes of Health [NIH] 1.34, http://rsbweb.nih.gov/ij/).

## In vitro tube formation assay

Tube formation assay was performed as described previously (*Yin, 2022*). MCPs were infected with lentiviruses ORF control particles (NC, $5 \times 10^5$ infection units per milliliter cultured medium; Origene Technology) or ORF clone of mouse Lbh (Lbh O/E, $5 \times 10^5$ infection units per milliliter cultured medium; Origene Technology) under HG conditions for at least 3 days. Tube formation assays were performed in a 48-well plate with 100 µL of growth factor-reduced Matrigel (Becton Dickinson). The assay was performed in a $CO_2$ incubator and the images were obtained at 24 hr with a screen magnification of 40 using a phase-contrast microscope (CKX41, Olympus, Japan). The numbers of master junctions from four separate experiments were determined using ImageJ software.

## Ex vivo neurite sprouting assay

MPG tissues were harvested and maintained as described previously (*Ghatak et al., 2022*). After, the tissue was covered with Matrigel and incubated at 37°C for 10 min in a 5% $CO_2$ atmosphere, followed by incubation in 1.2 mL of complete Neurobasal Medium (Gibco) containing 0.5 nM GlutaMAX-I (Gibco) and 2% serum-free B-27 (Gibco). The MPG tissues were infected with lentiviruses ORF control particles (NC, $5 \times 10^4$ infection units per milliliter cultured medium; Origene Technology) or ORF clone of mouse *Lbh* (LBH O/E, $5 \times 10^4$ infection units per milliliter cultured medium; Origene Technology) under NG or HG conditions. Five days later, neurite outgrowth segments were then fixed in 4% para-formaldehyde for at least 30 min and IF staining with an anti-neurofilament antibody (N4142; Sigma-Aldrich; 1:50). Images were obtained with a phase-contrast microscope (CKX41, Olympus, Japan). Quantitative analysis of neurite length was determined using ImageJ software.

## Proteome profiler mouse angiogenesis array analysis

Secreted angiogenesis factors in cavernous tissue between normal and diabetes condition were detected using a proteome profiler mouse angiogenesis array kit (ARY015; R&D Systems Inc), as described by the manufacturer. This array detects 53 mouse angiogenesis-related protein simultaneously. The intensity of dot blots was analyzed using ImageJ software. The criteria for selecting spots were an expression density of at least 1500 and a change ratio greater than 1.2, as shown in the supplementary information.

## Histological examination

For fluorescence examinations, tissues were fixed in 4% paraformaldehyde overnight at 4°C, and cell samples were fixed in 4% paraformaldehyde for 15 min at RT. After blocking with 1% BSA (Sigma-Aldrich) for 1 hr at RT, frozen tissue sections (12 µm as thick) or cell samples were incubated with primary antibodies at 4°C overnight. The antibodies used were as follows: anti-LBH (1:200; Novus Biologicals, Littleton, CO, USA), anti-CD31 antibody (1:50; Millipore), anti-PDGFRβ (1:100; Invitrogen), anti-α-SMA (1:100; Abcam, Cambridge, MA, USA), anti-CD140b (1:100; Invitrogen), anti-NG2 antibody (1:100; Millipore), anti-Neurofilament (1:100; Sigma-Aldrich), nNOS (1:50; Santa Cruz Biotechnology, Dallas, TX, USA), anti-phospho-HistoneH3 (1:50; Millipore; 1:50), anti-Vimentin (1:100; Sigma-Aldrich), and anti-Crystallin Alpha B (CRYAB, 1:100; Invitrogen). Wash the samples with PBS

(Gibco) for at least three times, tissue sections or cell samples were incubated with donkey anti-rabbit DyLight 550 (1:200; Abcam), donkey anti-mouse Alexa Fluor 488 (1:200; Jackson ImmunoResearch Laboratories, West Grove, PA, USA), goat anti-Armenian hamster Fluorescein (FITC) (1:200; Jackson ImmunoResearch Laboratories), and donkey anti-chicken rhodamine (TRITC) secondary antibodies (1:200; Jackson ImmunoResearch Laboratories) for 2 hr at RT. Samples were mounted with a solution containing 4,6-diamidino-2-phenylindole (H-1200, Vector Laboratories Inc, Burlingame, CA, USA) for nuclei staining. All images were obtained using a confocal microscope (K1-Fluo; Nanoscope Systems, Inc). Quantitative analysis was performed using ImageJ software (NIH 1.34, http://rsbweb.nih.gov/ij/).

### LC-MS/MS analysis of immunoprecipitates

The LC-MS/MS analysis was performed as a custom service by Yonsei Proteome Research Center (Yonsei Proteome Research Center, Seoul, Republic of Korea) as previously described (*Yin et al., 2022*). In brief, total cell lysates were immunoprecipitated with LBH antibody (1:50; Sigma-Aldrich), and analyzed by SDS-PAGE and Coomassie blue staining. The indicated bands (*Figure 6*, framed in red dot line) were excised from SDS-PAGE gels, and nano LC-MS/MS analyses were performed using an Easy n-LC (Thermo Fisher, San Jose, CA, USA) and an LTQ Orbitrap XL mass spectrometer (Thermo Fisher) equipped with a nano-electrospray source (*Lee et al., 2014*). Samples were separated on a C18 nanobore column (150 mm × 0.1 mm, 3 µm pore size; Agilent). Mobile phase A for LC separation was 0.1% formic acid plus 3% acetonitrile in deionized water and mobile phase B was 0.1% formic acid in acetonitrile. The chromatography gradient was designed to achieve a linear increase from 0% B to 32% B in 23 min, 32% B to 60% B in 3 min, 95% B in 3 min, and 0% B in 6 min. The flow rate was maintained at 1500 nL/min. Mass spectra were acquired using data-dependent acquisition with a full mass scan (350–1800 m/z) followed by 10 MS/MS scans. For MS1 full scans, the orbitrap resolution was 15,000 and the AGC was $2\times10^5$. For MS/MS in the LTQ, the AGC was $1\times10^4$. The Mascot algorithm (Matrix Science, USA) was used to identify peptide sequences present in a protein sequence database. Database search criteria were as follows: taxonomy, *Homo sapiens*, *M. musculus*; fixed modification, carbamidomethylated at cysteine residues; variable modification, oxidized at methionine residues; maximum allowed missed cleavages, 2; MS tolerance, 10 ppm; MS/MS tolerance, 0.8 Da. Only peptides resulting from trypsin digests were considered. Peptides were filtered with a significance threshold of $p<0.05$.

### Immunoblots and immunoprecipitation

Cells and tissues were lysed in RIPA buffer (Sigma-Aldrich) supplemented with protease (GenDEPOT, LLC, Katy, TX, USA) and phosphatase (GenDEPOT, LLC) inhibitors. Equal amounts of proteins (30 µg per lane) from whole-cell or tissue lysates were resolved by SDS-PAGE on 8–15% gels, and then transferred to polyvinylidene fluoride membranes. After blocking with 5% non-fat dried milk for 1 hr at RT, membranes were incubated at 4°C overnight with the following primary antibodies: anti-LBH (1:1000; Sigma-Aldrich), anti-Vimentin (1:1000; Sigma-Aldrich), and anti-CRYAB (1:1000; Invitrogen). For immunoprecipitation, 10% (50 µg) of lysate was used as positive control (input), 500 µg of lysate was incubated with the indicated antibody (1–2 µg) for 3–4 hr at 4°C followed by overnight incubation with Protein A/G PLUS-Agarose (Santa Cruz Biotechnology). Immunoprecipitates were washed five times with RIPA buffer, and then were resolved by SDS-PAGE and immunoblotted with the indicated antibodies. Densitometric analyses of western blot bands were performed using ImageJ software (NIH 1.34, http://rsbweb.nih.gov/ij/).

### Statistical analysis

Results are expressed as the means ± SEMs of at least four independent experiments. The unpaired t test was used to compare two groups, and one-way ANOVA followed by Tukey's post hoc test for four-group comparisons. The analysis was conducted using GraphPad Prism version 8 (GraphPad Software, Inc, La Jolla, CA, USA, https://graphpad.com), and statistical significance was accepted for p-values<0.05.

### Acknowledgements

This work was funded by the National Research Foundation of Korea (NRF) grants (Guo Nan Yin, NRF-2021R1A2C4002133, Ji-Kan Ryu, 2022R1A2B5B02001671, Jihwan Park, 2019R1C1C1005403 and

2021M3H9A2097520), and a Medical Research Center grant (Ji-Kan Ryu, NRF-2021R1A5A2031612) funded by the Korean government. This work was also supported by the 2023 GIST-MIT Research Collaboration grant funded by the GIST.

## Additional information

### Competing interests
Jihwan Park: Reviewing editor, *eLife*. The other authors declare that no competing interests exist.

### Funding

| Funder | Grant reference number | Author |
|---|---|---|
| National Research Foundation of Korea | 2021R1A2C4002133 | Guo Nan Yin |
| National Research Foundation of Korea | 2022R1A2B5B02001671 | Ji-Kan Ryu |
| National Research Foundation of Korea | 2019R1C1C1005403 | Jihwan Park |
| National Research Foundation of Korea | 2021M3H9A2097520 | Jihwan Park |
| Medical research center | 2021R1A5A2031612 | Ji-Kan Ryu |
| GIST | 2023 GIST-MIT Research Collaboration grant | Jihwan Park |

The funders had no role in study design, data collection and interpretation, or the decision to submit the work for publication.

### Author contributions
Seo-Gyeong Bae, Data curation, Formal analysis, Investigation, Visualization, Writing – original draft, Writing – review and editing; Guo Nan Yin, Conceptualization, Resources, Data curation, Funding acquisition, Validation, Investigation, Visualization, Writing – original draft, Writing – review and editing; Jiyeon Ock, Validation, Writing – review and editing; Jun-Kyu Suh, Conceptualization, Resources, Supervision, Project administration, Writing – review and editing; Ji-Kan Ryu, Conceptualization, Resources, Supervision, Funding acquisition, Project administration, Writing – review and editing; Jihwan Park, Conceptualization, Supervision, Funding acquisition, Project administration, Writing – review and editing

### Author ORCIDs
Seo-Gyeong Bae ⓘ http://orcid.org/0000-0001-8080-3783
Guo Nan Yin ⓘ https://orcid.org/0000-0002-2512-7337
Jihwan Park ⓘ http://orcid.org/0000-0002-5728-912X

### Ethics
Animal experiments were approved by the Institutional Animal Care and Use Subcommittee of Inha university (approval number: 200910-719).

Reviewer #1 (Public Review): https://doi.org/10.7554/eLife.88942.4.sa1
Reviewer #3 (Public Review): https://doi.org/10.7554/eLife.88942.4.sa2
Author response https://doi.org/10.7554/eLife.88942.4.sa3

## Additional files

### Supplementary files
• Supplementary file 1. Relative density of mouse angiogenesis array spots.

• Supplementary file 2. Physiological and metabolic parameters: 2 weeks after treatment with PBS, NC, LBH O/E. Values are the mean ± SEMs for n=5 animals for each group. STZ, streptozotocin; MSBP, mean systolic blood pressure; NC, lentiviruses ORF control particles; LBH O/E, ORF clone of mouse Lbh lentiviruses, *p<0.05 vs. control group.

• MDAR checklist

## Data availability

The raw data was deposited in Korean Nucleotide Archive (KoNA, https://kobic.re.kr/kona) with the accession ID, KAP230548. All data associated with this study are available in the main text or the supplementary materials.

The following dataset was generated:

| Author(s) | Year | Dataset title | Dataset URL | Database and Identifier |
|---|---|---|---|---|
| Bae SG, Yin GN, Ock J, Suh J, Ryu J, Park J | 2023 | mouse penis single cell study | https://kobic.re.kr/kona/search_bioproject?bioproject_id=KAP230548 | Korean Nucleotide Archive, KAP230548 |

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
