## [Editor Report · eLife assessment]

The authors provide **important** insights into the pathogenesis of erectile dysfunction (ED) in patients with diabetes. The authors present **compelling** evidence, using single-cell transcriptomic analysis in both mouse and human cavernous tissues, to support their claims regarding the key roles of pericytes in diabetic ED. The identification of LBH as a potential pericyte-specific marker in both mouse and human tissues further strengthens their findings. This well-written manuscript offers novel and significant contributions to the field, identifying potential therapeutic targets for further investigation.

---

## [Referee Report · Reviewer #1 (Public Review)]

In this study, the researchers aimed to investigate the cellular landscape and cell-cell interactions in cavernous tissues under diabetic conditions, specifically focusing on erectile dysfunction (ED). They employed single-cell RNA sequencing to analyze gene expression patterns in various cell types within the cavernous tissues of diabetic individuals. The researchers identified decreased expression of genes associated with collagen or extracellular matrix organization and angiogenesis in several cell types, including fibroblasts, chondrocytes, myofibroblasts, valve-related lymphatic endothelial cells, and pericytes. They also discovered a newly identified marker, LBH, that distinguishes pericytes from smooth muscle cells in mouse and human cavernous tissues. Furthermore, the study revealed that pericytes play a role in angiogenesis, adhesion, and migration by communicating with other cell types within the corpus cavernosum. However, these interactions were found to be significantly reduced under diabetic conditions. The study also investigated the role of LBH and its interactions with other proteins (CRYAB and VIM) in maintaining pericyte function and highlighted their potential involvement in regulating neurovascular regeneration. Overall, the manuscript is well-written and the study provides novel insights into the pathogenesis of ED in patients with diabetes and identifies potential therapeutic targets for further investigation.

Comments on revised version:

All my concerns have been properly addressed.

---

## [Referee Report · Reviewer #3 (Public Review)]

Bae and colleagues substantially improved the data quality and revised their manuscript "Single cell transcriptome analysis of cavernous tissues reveals the key roles of pericytes in diabetic erectile dysfunction". While these revisions clarify some of the concerns raised, others remain. In my view, the following question must be addressed:

In my prior question on #3, I completely disagree with the statement that "identified cells with pericyte-like characteristics in the walls of large blood vessels". The staining that authors provided for LBH, was clearly stained for SMCs, not pericytes. Per Fig 2E, the authors are correct that LBH is colocalized with SMA+ cells( SMCs). However, the red signal from LBH clearly stains endothelial cells. In the rest of 2E and 2D, LBH is CD31- and their location suggests LBH stained for SMCs in the Aorta, Kidney vasculature, Dorsal vein, and Dorsal Artery.

---

## [Author Response]

The following is the authors’ response to the previous reviews.

**eLife assessment**
The authors have made important contributions to our understanding of the pathogenesis of erectile dysfunction (ED) in diabetic patients. They have identified the gene Lbh, expressed in pericytes of the penis and decreased in diabetic animals. Overexpression of Lbh appears to counteract ED in these animals. The authors also confirm Lbh as a potential marker in cavernous tissues in both humans and mice. While solid evidence supports Lbh's functional role as a marker gene, further research is needed to elucidate the specific mechanisms by which it exerts its effects. This work is of interest to those working in the fields of ED and angiogenesis.
**Public Reviews:**

**Reviewer #1 (Public Review):**
In this study, the researchers aimed to investigate the cellular landscape and cell-cell interactions in cavernous tissues under diabetic conditions, specifically focusing on erectile dysfunction (ED). They employed single-cell RNA sequencing to analyze gene expression patterns in various cell types within the cavernous tissues of diabetic individuals. The researchers identified decreased expression of genes associated with collagen or extracellular matrix organization and angiogenesis in several cell types, including fibroblasts, chondrocytes, myofibroblasts, valve-related lymphatic endothelial cells, and pericytes. They also discovered a newly identified marker, LBH, that distinguishes pericytes from smooth muscle cells in mouse and human cavernous tissues. Furthermore, the study revealed that pericytes play a role in angiogenesis, adhesion, and migration by communicating with other cell types within the corpus cavernosum. However, these interactions were found to be significantly reduced under diabetic conditions. The study also investigated the role of LBH and its interactions with other proteins (CRYAB and VIM) in maintaining pericyte function and highlighted their potential involvement in regulating neurovascular regeneration. Overall, the manuscript is well-written and the study provides novel insights into the pathogenesis of ED in patients with diabetes and identifies potential therapeutic targets for further investigation.Comments on revised version:For Figure 4, immunofluorecent staining of LBH following intracavernous injections with lentiviruses is required to justify overexpression and tissue specificity.

We agree with this claims. Therefore, we have performed the immunofluorecent staining of LBH in cavernous tissues after infection with LBH O/E lentiviruses. And we found the LBH expression is significantly decreased in DM or DM+NC groups, however, after infection with LBH O/E lentiviruses, the LBH expression is significantly increased, shown as Supplementary Fig. 10. (Please see revised ‘Result’ and ‘Supplementary Fig. 10’)

**Reviewer #3 (Public Review):**
Bae et al. described the key roles of pericytes in cavernous tissues in diabetic erectile dysfunction using both mouse and human single-cell transcriptomic analysis. Erectile dysfunction (ED) is caused by dysfunction of the cavernous tissue and affects a significant proportion of men aged 40-70. The most common treatment for ED is phosphodiesterase 5 inhibitors; however, these are less effective in patients with diabetic ED. Therefore, there is an unmet need for a better understanding of the cavernous microenvironment, cell-cell communications in patients with diabetic ED, and the development of new therapeutic treatments to improve the quality of life.Pericytes are mesenchymal-derived mural cells that directly interact with capillary endothelial cells (ECs). They play a vital role in the pathogenesis of erectile function as their interactions with ECs are essential for penile erection. Loss of pericytes has been associated with diabetic retinopathy, cancer, and Alzheimer's disease and has been investigated in relation to the permeability of cavernous blood vessels and neurovascular regeneration in the authors' previous studies. This manuscript explores the mechanisms underlying the effect of diabetes on pericyte dysfunction in ED. Additionally, the cellular landscape of cavernous tissues and cell type-specific transcriptional changes were carefully examined using both mouse and human single-cell RNA sequencing in diabetic ED. The novelty of this work lies in the identification of a newly identified pericyte (PC)-specific marker, LBH, in mouse and human cavernous tissues, which distinguishes pericytes from smooth muscle cells. LBH not only serves as a cavernous pericyte marker, but its expression level is also reduced in diabetic conditions. The LBH-interacting proteins (Cryab and Vim) were further identified in mouse cavernous pericytes, indicating that these signaling interactions are critical for maintaining normal pericyte function. Overall, this study demonstrates the novel marker of pericytes and highlights the critical role of pericytes in diabetic ED.Comments on revised version:Bae and colleagues substantially improved the data quality and revised their manuscript "Pericytes contribute to pulmonary vascular remodeling via HIF2a signaling". While these revisions clarify some of the concerns raised, others remain. In my view, the following question must be addressed.In my prior question on #3, I completely disagree with the statement that "identified cells with pericyte-like characteristics in the walls of large blood vessels". The staining that authors provided for LBH, was clearly stained for SMCs, not pericytes. Per Fig 2E, the authors are correct that LBH is colocalized with SMA+ cells( SMCs). However, the red signal from LBH clearly stains endothelial cells. In the rest of 2E and 2D, LBH is CD31- and their location suggests LBH stained for SMCs in the Aorta, Kidney vasculature, Dorsal vein, and Dorsal Artery.

We respect the reviewer's comments and provide further justification for the reviewer's concerns. We first performed double staining of LBH and CD31 on dorsal artery and dorsal vein tissues. We found that LBH-expressing cells are completely different from CD31-expressing cells (Figrue 2D, indicated by arrows, and Supplementary Fig. 10A) and that expression is higher in veins than in arteries. This is consistent with previous understanding. In addition, in the double staining of LBH and α-SMA, we also found that there was no overlap between LBH-expressing cells and α-SMA-expressing smooth muscle cells in the cavernosum tissues, but there was some overlap in dorsal artery and dorsal vein (Figrue 2E, indicated by arrows). This may indicate that LBH is expressed slightly different types of blood vessels. This requires further experiments to prove in the future. In addition, to avoid confusion among other readers. We modify our previous discussion regarding the identification of cells with pericyte-like characteristics in the walls of large blood vessels. We removed the associated immunofluorescence staining in the aorta and kidneys replaced them with dorsal artery and dorsal vein (Please see revised ‘Result’ and ‘Figure 2’ and ‘Supplementary Fig. 10A’)